# Nutraceuticals for Gut–Brain Axis Health: A Novel Approach to Combat Malnutrition and Future Personalised Nutraceutical Interventions

**DOI:** 10.3390/nu17091551

**Published:** 2025-04-30

**Authors:** Litai Liu, Wen Qi, Na Zhang, Jinhao Zhang, Shen Liu, Huan Wang, Lianzhou Jiang, Ying Sun

**Affiliations:** 1Tourism & Cuisine College, Harbin University of Commerce, Harbin 150028, China; tailau321@yahoo.com (L.L.); foodqiwen@163.com (W.Q.); foodzhangna@163.com (N.Z.); 13223007406@163.com (S.L.); 2Department of Food and Nutritional Sciences, University of Reading, Reading RG6 6UR, UK; 3College of Food Science, Northeast Agricultural University, Harbin 150030, China; foodjinhao@163.com (J.Z.); whname@neau.edu.cn (H.W.); jlzname@163.com (L.J.)

**Keywords:** malnutrition, nutraceutical, gut microbiota, gut–brain axis, personalised nutraceuticals

## Abstract

The gut–brain axis (GBA) is a bidirectional communication network between the gastrointestinal tract and the brain, modulated by gut microbiota and related biomarkers. Malnutrition disrupts GBA homeostasis, exacerbating GBA dysfunction through gut dysbiosis, impaired neuroactive metabolite production, and systemic inflammation. Nutraceuticals, including probiotics, prebiotics, synbiotics, postbiotics, and paraprobiotics, offer a promising approach to improving GBA homeostasis by modulating the gut microbiota composition and related neuroactive metabolites. This review aims to elucidate the interplay between gut microbiota-derived biomarkers and GBA dysfunction in malnutrition and evaluate the potential of nutraceuticals in combating malnutrition. Furthermore, it explores the future of personalised nutraceutical interventions tailored to individual genetic and microbiome profiles, providing a targeted approach to optimise health outcomes. The integration of nutraceuticals into GBA health management could transform malnutrition treatment and improve cognitive and metabolic health.

## 1. Introduction

The complex relationship between nutrition, gut health, and brain function has long been recognised: the gut–brain axis (GBA) is a bidirectional communication network, and this axis is significantly influenced by the gut microbiota, a diverse community of microorganisms that reside within the gastrointestinal tract (GIT) and play a crucial role in modulating host metabolism, energy homeostasis, immune function, and eating behaviour [1]. Accumulating evidence suggests that the gut microbiota can produce neurotransmitters (e.g., serotonin, dopamine, and norepinephrine) and short chain fatty acids (SCFAs) such as acetate, butyrate, and propionate, which affect inflammation and appetite regulation, as well as affect cognitive and emotional states, highlighting their potential as a therapeutic target for various health conditions [2,3]. Disruptions in the GBA, often driven by factors such as poor diet, stress, or disease, have been implicated in a wide range of health conditions, including malnutrition, mental health disorders, and metabolic diseases [4]. The prevalence of several chronic diseases related to nutritional status continues to increase worldwide, Malnutrition is a condition that occurs when an individual’s dietary intake does not contain the appropriate amount of nutrients to maintain good health, leading to malnutrition. Malnutrition is defined as excessive or insufficient nutrient intake, an imbalance of essential nutrients, or impaired utilisation of nutrients, as seen in Table 1.

The double burden of malnutrition includes obesity/overweight and overnutrition [5]. The worldwide impact of malnutrition includes various nutritional issues like undernutrition, overnutrition, and micronutrient deficiencies, all of which pose major public health concerns. In 2022, approximately 2.5 billion adults were overweight, with 890 million classified as obese and 390 million underweight. Among children under five, 149 million were stunted, 45 million were wasted, and 37 million were overweight or obese [5]. Malnutrition effects are not only associated with physical health consequences but also have profound impacts on the nervous system, leading to cognitive impairments, mood disorders, and developmental delays [9]. For example, unbalanced intake of essential nutrients can lead to impaired gut barrier function, weakened immune function, and altered levels of neurotransmitters (NTs) and appetite hormones, ultimately affecting brain health and cognitive performance. and further exacerbating GBA dysregulation. Hence, combating malnutrition and re-establishing equilibrium in the GBA are vital for optimising health outcomes and mitigating associated comorbidities.

The prevalence of malnutrition and its neuropsychological impacts highlights the need for effective therapeutic strategies. Nutraceuticals represent a particularly promising approach compared to conventional therapeutic strategies due to their multiple benefits and minimal side effects. Compared to pharmaceutical drugs, nutraceuticals are derived from natural sources and often have a lower risk of adverse effects. Moreover, nutraceuticals can target multiple pathways such as modulating gut microbiota composition and its metabolites, reducing inflammation, and enhancing gut barrier integrity, which are all crucial for improving GBA health [10]. Nutraceuticals, which are food-derived compounds with bioactive properties, have emerged as a promising strategy for modulating the GBA and combating malnutrition [11]. Nutraceuticals, including probiotics, prebiotics, and synbiotics, have been shown to modulate gut microbiota composition, enhance gut barrier integrity, decrease systemic inflammation, and regulate neurotransmitter and appetite hormones [12,13,14,15,16,17,18]. These nutraceuticals, often available through dietary supplements, can directly modulate the microbiome, providing therapeutic benefits for both gut and brain health [19]. Advances in personalised medicine have paved the way for tailored nutraceutical interventions based on individual genetic and microbiome profiles. Personalised nutraceutical interventions that take into account the unique characteristics of individuals’ gut microbiota and genetic makeup provide a more targeted and effective approach to optimising health outcomes. As research continues to uncover the complex interplay between the gut microbiota and the brain, the potential of nutraceuticals to improve GBA health and combat malnutrition becomes more apparent [20]. This review addresses three central questions: (1) How do malnutrition-induced disruptions in the GBA drive neurological and metabolic dysfunction? (2) What is the impact of nutraceuticals on gut microbiota composition and GBA-related biomarkers in the context of malnutrition? (3) What are the prospects and challenges for developing personalised nutraceutical interventions based on individual genetic and microbiome profiles? To answer these questions, this review aims to explore the mechanisms underlying the GBA in malnutrition, the potential of nutraceuticals in improving gut microbiota composition and GBA-related biomarkers, and the prospects for personalised nutraceutical interventions. By highlighting the research findings and identifying gaps in current knowledge, this review seeks to provide insights into the future directions of tailored nutraceutical research and its potential applications in addressing malnutrition.

## 2. Combatting Malnutrition via Various GBA Biomarkers

The gut microbiota is an intricate ecosystem comprising trillions of microorganisms that influence both the host’s normal physiological functions and its predisposition to disease [2]. It consists of over 100 bacterial species, with a gene count 150 times greater than that of the human genome [21,22]. Besides bacteria, the gut microbiota includes protozoa, archaea, viruses, and fungi [23]. The gut microbiota composition is dynamic, changing in response to environmental factors like diet and host factors such as age and genetics [24]; it performs various functions, including maintaining gut mucosal integrity, stimulating mucus production, and facilitating SCFA synthesis [25]. Additionally, the gut microbiota supports the maturation of the innate immune system during early life and processes numerous environmental signals to influence overall health. Gut microbiota acting as a bridge between the host and the environment and changes in the gut microbiota can have profound effects on human health [26]. Alterations in beneficial bacteria can impact on human health and potentially trigger certain disease mechanisms. Diet, illness, medications, and infections are factors that can modify the gut microbiota composition and function [27,28]. GBA is defined as bidirectional communication between gut microbiota and the brain through multiple systems that form a network. It plays an important role in maintaining homeostasis in the central nervous system (CNS) and GIT [27]. The interacting pathways in this network include direct and indirect signalling through neuronal pathways, the immune system, appetite hormones, and chemical neurotransmitters. Since these networks involve different biological systems, their complexity needs to be investigated [26].

The CNS continually reacts to a range of chemical and neural signals that track the individual’s energy status. These signals include NTs, appetite hormones, and SCFAs, which are produced primarily in the GIT [29]. GBA is influenced by a range of factors such as diet, gut microbiota, and genetics [30]. However, the specific pathways through which the gut microbiota affect appetite regulation via the brain remain unclear. Accumulated studies have highlighted the critical role of the gut microbiota in the regulation of multiple biomarkers, including appetite hormones, NTs, immune markers, and SCFAs; these biomarkers are interlinked and play a role in the development and progression of malnutrition, as illustrated in Figure 1. Utilising these biomarkers can be an effective approach to tackle malnutrition, offering insights into nutritional status and guiding targeted interventions [31,32].

### 2.1. Gut Microbiota Effects on SCFAs Regulation

The most abundant SCFAs detected in the faecal are acetate, butyrate, and propionate within the GIT [33]. Butyrate is mainly produced by Firmicutes including Lachnospiraceae and *Faecalibacterium prausnitzii*, while propionate is primarily produced by *Bacteroides* species, *Clostridium* species, and Negativicutes [34]. SCFAs are essential for modulating the immune system and maintaining intestinal homeostasis. For example, SCFAs exhibit anti-inflammatory effects by modulating immune cell activity, reducing pro-inflammatory cytokine release, and strengthening the integrity of the intestinal barrier to prevent the translocation of endotoxins [35]. Neuroinflammation is a key factor when considering brain function. Studies have shown that disturbances in the gut microbiota, such as those induced by antibiotics, lead to systemic immune dysregulation, characterised by pro-inflammatory profiles [35]. In the CNS, antibiotic-induced depletion of the microbiota activates an inflammatory response that alters microglial morphology. Microglia, as the resident immune cells of the CNS, are responsive to changes in the gut microbiota composition and its metabolites [36]. SCFAs, particularly butyrate, have emerged as key mediators linking gut microbiota to neuroinflammation [37]. Sodium butyrate has been shown to inhibit microglial activation and the secretion of pro-inflammatory cytokines, thereby reducing neuroinflammation. For instance, studies have demonstrated that sodium butyrate can inhibit lipopolysaccharide-induced depression-like symptoms in mice by modulating microglial activation [38]. Additionally, butyrate has been shown to promote the differentiation of regulatory T cells and reduce pro-inflammatory cytokines production, further highlighting its anti-inflammatory potential properties [35]. Moreover, SCFAs can affect the GBA axis through multiple mechanisms, including modulation of the vagus nerve and alteration of gut barrier function, and direct effects on NTs production. These mechanisms contribute to neuroinflammation regulation and may provide new insights into the treatment of neurodegenerative and neurodevelopmental disorders [39].

### 2.2. Immune Markers Effect on Gut Microbiota and NTs Regulation

Accumulated animal studies emphasised that a high-fat diet leads to alternations in gut microbiota composition and enhanced intestinal permeability [40]. In a murine study, a high-fat diet led to higher Firmicutes phylum and lower Bacteroides in faeces, along with higher pro-inflammatory cytokines such as tumour necrosis factor alpha (TNF-α), interleukin-1 beta (IL-1β), and interleukin-6 (IL-6) in plasma [41]. Currently, there is no research involving immune–neurotransmitter interaction in individuals with malnutrition. However, as diet can impact on NTs, high-fat-induced inflammation leads to negative effects on cognition and behaviour via dysregulated neurotransmission [42]. As such, changes in dietary regimes are likely, through the gut microbiota, to impact on the brain.

### 2.3. Gut Microbiota Effects on Appetite Hormones Regulation

Gut microbiota play a key role in host appetite control via gut hormone regulation [43]. Enterobacteria such as *Escherichia coli* are prominent residents of the gut microbiota and are capable of producing small protein sequences, caseinolytic protease B (CIpB) [43,44]. CIpB is a conformational antigen mimic of α-MSH that seems to induce α-MSH and activate anorexigenic brain neurons involved in anxiety and satiety signals [45,46]. For example, a mouse study showed that mice immunised with CIpB bacterial protein had decreased food consumption and bodyweight and increased anxiety [47]. CIpB has been observed to be increased in the plasma of patients with eating disorders associated with insufficient food intake, such as AN. Additionally, ghrelin has been implicated in the regulation of food intake and energy homeostasis in mammals [48]. Galactooligosaccharides (GOS) are enzymatically produced from lactose by the food industry. They are widely used in infant nutrition formulations to mimic the biological functions of human milk oligosaccharides (HMOs), such as effects on gut microbiota and the immune system [49]. GOS escapes digestion and absorption due to lack of the appropriate digestive enzymes in the small intestine. After arriving in the colon, GOS are metabolised by resident microbial species [50]. A study confirmed that GOS-fed mice displayed increased gene expression of satiety-related peptides, so it was observed that gut microbiota regulated by prebiotics had enhanced genes of glucagon-like peptide (GLP-1) precursor proglucagon expression by 3.5-fold and 1.5-fold of peptide YY (PYY) in the colonic mucosal [51]. Accumulating evidence suggests that the gut microbiota composition can help in regulation of appetite hormones.

### 2.4. Gut Microbiota Effects on NTs Regulation

The gut microbiota has been related to the production of not only SCFAs but also gut microbial-derived NTs, including serotonin (5-HT), gamma aminobutyric acid (GABA), dopamine (DA), norepinephrine (NE), and epinephrine [52,53]. There have been studies on the potential role of the gut microbiome on host NTs and their related pathways with outcomes for behaviour and host physiology. For instance, compared to specific pathogen-free mice, germ-free mice have decreased 5-HT receptors and circulating 5-HT in the hippocampus, and this is accompanied by altered anxiety-like behaviour [53]. Some studies have indicated altered concentrations of NTs in germ-free mice following supplementation with defined gut bacteria [54]. DA and 5-HT are reported to be produced by several gut bacteria, and this is likely to have an impact on the brain, as the total concentration of tryptophan, glutamine, and tyrosine in the brain of germ-free mice is lower than the mice with gut microbiota recolonisation [55], whilst strains of *Bifidobacterium* and *Lactobacillus* can be regarded as natural residents of the intestine with beneficial impacts on GABA production [56,57]. The gut microbiota is considered to be a modulator of NT levels, which then operate through the GBA (vagus nerve pathway). The variation in gut microbiota-associated communication in the gut–microbiota–brain axis has been implicated in aspects of physiological and psychological conditions, including neurologic, immunologic and psychiatric conditions [58,59]. For example, variation in the gut microbial community has been reported in CNS disorders, including autistic spectrum disorders, anxiety, and depression [60]. Neurological diseases such as depression (decreased 5-HT and catecholamines), Parkinson’s (decreased DA), insomnia, and anxiety (decreased GABA) are linked to deficits in certain NTs [61,62]. The changes observed in mental well-being may therefore be caused by signal transduction from the intestine to the brain [63].

### 2.5. SCFAs Effect on Immune Markers Regulation

A study has reported that patients with malnutritional status have increased pro-inflammatory cytokines levels in plasma; these are IL-6, interleukin 17 (IL-17), and TNF-α when compared to healthy controls [64]. SCFAs, especially butyrate, as the main energy source of colonocytes, supports gut barrier function and exerts anti-inflammatory effects [65]. For example, TNF-α production induced by stimuli in vitro could be suppressed by acetate and butyrate [66]. An in vitro study indicated that butyrate could inhibit proinflammatory cytokines by restricting lipopolysaccharide-induced nuclear factor-κB activity [66,67]. 

### 2.6. SCFAs Effect on Appetite Hormones Regulation

SCFAs are involved in regulating the expression of appetite hormones and energy homeostasis [68]. It has been reported that propionate stimulates both PYY and GLP-1 secretions from wild-type primary murine colonic crypt cultures [69]. As such, there are communication pathways among the gut microbiome, SCFAs, and anorexigenic/orexigenic hormones, which could have potential for normalising the satiety hormone levels in NDs and therefore have a potential role in therapeutic feeding regimes. Propionate is an energy source for the epithelial cells, also transferred to the liver, where it plays a role in gluconeogenesis and thus is considered beneficial for glucose and energy homeostasis. A murine study indicated that the higher physiological levels of propionate significantly stimulated anorexigenic hormones, including GLP-1 and PYY [68], suggesting that increased levels of SCFAs would stimulate gut hormone profiles thus impacting on appetite.

### 2.7. SCFAs Effect on NTs Regulation

Gut microbiota acting through SCFAs can upregulate enteric 5-HT production and homeostasis by the enterochromaffin cells (ECs) [70]. 5-HT synthesis is regulated by tryptophan hydroxylase that is a rate-limiting enzyme that participates in 5-HT synthesis [61]. A study showed both mouse- and human-derived gut microbiota promote colonic tryptophan hydroxylase expression and 5-HT amounts through SCFA activities on ECs [68]. Also, SCFAs have been observed to increase DA synthesis through tyrosine hydroxylase, a major enzyme in catecholamine synthesis [71].

### 2.8. Appetite Hormones Effect on Immune Markers Regulation

There is still limited research focusing on appetite hormone effects on immune markers. Sepsis has been observed to reduce levels of ghrelin, as such a rat study performed caecal ligation on males to administer sepsis. This is followed by an injection of ghrelin. Ghrelin infusion restored brain levels and alleviated intestinal barrier dysfunction with highly active group B1 serum levels and vagus nerves [72]. Therefore, satiety hormones may also impact on gut barrier function and subsequently immune function. This closely linked network is likely to impact on many illnesses, and malnutrition is a good target, as malnutrition conditions are highly linked to diet.

### 2.9. NTs Effect in Appetite Hormone Regulation

Accumulated studies have indicated that NTs participate in hypothalamic appetite regulation [73]. DA and 5-HT are essential NTs in the regulation of food intake; alteration in amounts of DA and 5-HT in the ventromedial nucleus and lateral hypothalamic area relate to the impact on food choice, including meal size and meal number [74]. Levels of NE within the brain may directly alter the leptin activity. There appears to be an inverse relationship between NE and leptin activity. Enhanced leptin may decrease NE activity to induce satiety, whereas the absence of leptin may stimulate increased NE secretion and subsequently release hunger signals [75].

### 2.10. Malnutrition and GBA Dysregulation

A few systematic reviews have demonstrated that individuals with malnutrition exhibit varying habitual dietary intake patterns. For example, individuals with obese or overweight often have Western and high-fat diets [76]; patients with AN tend to follow a vegetarian diet, while undernourished children usually have a low dietary diversity intake [76,77]. Different habitual dietary intakes can have a major impact on the type of gut microbiota in an individual, as seen in Figure 2 [78,79,80]. Dietary habits can significantly influence the gut microbiota composition. Diets that are ultra-processed-food high in levels of calorie-dense, salt, processed carbohydrates, and saturated fat, and low in fibre, are known as the ‘Western diet’ and are linked to the Bacteroides predominant enterotype [81]. A vegetarian diet typically includes vegetables, fruits, nuts, legumes, and grains, possibly incorporate with eggs and dairy products, while excluding red meat consumption, which is connected to the predominantly *Prevotella* enterotype [78].

Hence, it can be seen that diet-induced malnutrition can lead to an imbalance in the gut microbiome, which can impact the GBA [82]. For example, approximately 90% of 5-HT is produced in the gut by enterochromaffin cells, with gut microbiota influencing its synthesis. Malnutrition decreases the intake of dietary tryptophan, which in turn reduces 5-HT production and availability. As tryptophan is a precursor for 5-HT synthesis, its reduced intake can lower brain 5-HT levels, and alterations in 5-HT signalling are not only associated with appetite regulation, but may also trigger eating disorders [83]. In cases of malnutrition, particularly protein-energy malnutrition, systemic inflammation may be triggered. This inflammation can impact the intestinal lining and increase its permeability, enabling harmful bacteria and toxins to infiltrate the bloodstream, which may subsequently affect brain function. Moreover, intestinal inflammation can activate brain signalling pathways that influence eating behaviour [84]. Accumulated studies have indicated that dysregulated levels of SCFAs, NTs, appetite hormones, and immune markers are associated with malnutritional status, as seen in Table 2. These GBA markers are components of the complex neural circuits that regulate GBA in malnutritional status. Understanding their roles and complex interactions can provide insights into the development of effective therapeutic interventions to address malnutrition [85].

## 3. Emerging Nutraceuticals for Treating Malnutrition via GBA Targeting

Nutraceuticals are characterised as foods or food ingredients that provide health benefits beyond essential nutrition, such as disease treatment and prevention [93]. Prebiotics, probiotics, synbiotics, postbiotics, and paraprobiotics are classified as nutraceuticals because they deliver health benefits that extend beyond fundamental nutrition and can be utilised to prevent or manage various health conditions [94]. The above sections indicate that individuals with malnutrition with a dietary nutrient imbalance may suffer from impairments to gut microbiota development that could result in dysbiosis of GBA biomarkers associated with metabolic and physiological conditions. Therefore, it is possible that prebiotics, probiotics, synbiotics, postbiotics, and paraprobiotics intervention may potentially help restore the gut microbiota composition and related levels of GBA biomarkers to combat dysregulated metabolic functions and offer some benefits against unbalanced nutritional status, in order to obtain better long-term clinical outcomes [12,13,14,15,16,17,18,95]. Although the impact of GBA and related biomarkers in individuals with malnutrition are still unclear, probiotics, prebiotics, synbiotics, postbiotics, and paraprobiotics serve as approaches to reestablish a healthy gut microbiota and modulate GBA biomarkers in individuals with malnutrition, as indicated in Table 3.

### 3.1. Probiotics

#### Effect of Probiotics on Gut Microbiota Composition and NTs Modulation

Probiotics are one of the most explored and utilised functional food ingredients with a variety of health-promoting properties [118]. Probiotics are defined as “live microorganisms that, when administered in adequate amounts, confer a health benefit on the host” [119]. Probiotics consist of *Bacillus* spp., *Escherichia coli*, yeasts (e.g., *Saccharomyces* spp.), and lactic acid bacteria (e.g., streptococci, lactobacilli, and bifidobacteria) [119]. It has now been established that probiotics can modulate the host gut microbiota in a beneficial manner [120]. For instance, it is suggested to use specific probiotics to restore the gut microbial balance by affecting the ratio of Firmicutes to Bacteroidetes (F/B) in malnutritional status [121]. However, it should be noted that relevant probiotic strains can differ significantly in terms of functional and structural concentrations, such as the species *Lactobacillus*. Several studies have shown that the administration of certain *Lactobacillus* species has different effects on the ratio of F/B in obese mice. Supplementation with species of *Lactobacillus paracasei* HII01 and *Lactobacillus reuteri* MM4–1A led to a decrease in the ratio of F/B [96,99], while the administration of *Lactobacillus brevis* DPC6108 and *Lactobacillus brevis* DSM32386 species resulted in an increase in the ratio of F/B [97]. Therefore, these differences emphasise the importance of carefully selecting and characterising probiotic strains to target different states of NDs, ensuring that the desired health benefits are achieved.

DA, 5-HT, and NE are categorised as monoaminergic NTs; research in both animals and human studies have shown that diet-induced starvation leads to a depletion of central monoamines, resulting in imbalanced NTs and altered receptor sensitivity [122]. Animal studies have demonstrated that both single-strain and mixed probiotic supplementation can significantly regulate body weight, as well as the levels of DA, NE, and 5-HT related to reward and appetite [100,103]. Moreover, in the in vitro model of AN under the intervention of *Saccharomyces boulardii*, the levels of DA and 5-HT were restored, showing improvement compared to the levels before intervention [109]. These changes suggest the potential of using probiotics to modulate the microbiota and also biomarkers related to the GBA, to affect the reward system and appetite regulation in the brain, which is essential for managing eating behaviours [108].

Malnutrition represents a systemic state of low-grade inflammation. Pro-inflammatory cytokines and other immune markers are tightly associated with the development of malnutritional status [123]. A lower inflammatory status is correlated with a lower risk of depression and better mood, which indirectly affects dysregulated eating behaviours [124]. Several studies have indicated that probiotics have a potential impact on immune regulation via the cytokine expression modulation. Three studies observed that obese mice fed with probiotics resulted in a lower profile of pro-inflammatory cytokines such as TNF-α and IL-6 [13,97,99]. Additionally, probiotics strengthen the gut barrier, preventing the leakage of inflammatory molecules into the bloodstream, which can disrupt metabolic processes and appetite regulation. For instance, two studies have demonstrated that supplementation with *Lactobacullis* in obese mice upregulates the expression of the tight junction protein ZOO-1, a protein that is significantly linked to inflammatory status [45,101]. In current preclinical and clinical studies, probiotics have been shown to affect the gut microbiome and metabolic activity, leading to neuroactive compounds production and affecting immune markers; these probiotics need to be further studied in individuals with malnutrition, specifically targeting their therapeutic efficacy, long-term impacts, and safety.

### 3.2. Prebiotics

#### Effect of Prebiotic Type on Neuroactive Metabolites

Prebiotics are defined as “substrates that are selectively utilised by host microorganisms conferring health benefits to the host” [125]. Prebiotics have a bifidogenic capability, specifically supplying a fermentable dietary source that can increase the growth of positive microorganisms such as *Bifidobacteria* and *Lactobacillus* [126]. Nonetheless, as the knowledge of diversity in gut microbiome expands, there are other target genera, including *Faecalibacterium*, *Roseburia*, *Akkermansia*, and *Propionibacterium* [127]. GOS and Fructo-oligosaccharides (FOS) are the main type of prebiotics and are the most extensively researched [128]. Several studies have indicated that prebiotic-based treatments reshape gut microbiota composition; this involves simulating the growth of *Bifidobacterium* and *Lactobacillus*, which has been observed in both obese human and mice supplemented with prebiotic oligofructose (OFS) [52,129]. Moreover, some research has shown that these alternations of gut microbiota composition were linked to enhanced entero-endocrine cell activity and appetite sensitivity [129]. For instance, these gut bacteria were associated with elevated levels of GLP-1 and PYY, as well as a suppressed ghrelin production and a notable reduction of appetite [130]. In fact, supplementing with OFS in a high-fat diet led to an increase in the number of *Bifidobacterium* in the gut and mitigated obesity-related symptoms such as a decrease in body weight and food intake [15,16].

FOS has been proven to inhibit the gut colonisation of pathogens, offering a protective benefit against both chronic and acute intestinal disorders [131]. In vitro study on individuals with AN showed that FOS feeding could potentially restore the bacterial community in the GIT, as well as major SCFAs metabolites such as acetate [109]. Additionally, Borgo et al. [132] assessed SCFA concentrations in plasma and found that acetate was the only detectable metabolite, indicating that is may be transported across the blood–brain barrier and may affect early brain development [133]. It should be noted that SCFAs serve as crucial metabolites in peripheral tissues, acting as a substrate for lipogenesis and influencing appetite regulation [134]. Indeed, acetate plays a role in modulating the expression of ghrelin [68]. Ghrelin is recognised for its appetite-stimulating hormone, and studies have shown that germ-free mice have significantly lower ghrelin levels than in conventional mice. The infusion of acetate led to increased caloric intake and ghrelin concentrations, suggesting that elevated acetate concentrations amplify ghrelin expression and thus appetite [135]. Therefore, there are interrelated pathways among the gut bacterial community, SCFAs, and orexigenic/anorexigenic hormones, which could potentially be utilised in therapeutic feeding strategies for individuals with malnutrition.

### 3.3. Synbiotics

Synbiotics are a combination of prebiotic and probiotic designed to hold a synergistic capacity [136]. This synergistic effect boosts the effectiveness and the survival of beneficial microorganisms in the gut. Thus, this is flexibility in the selection of live microorganisms and substrate for the determination of the optimal combination for a specific desired outcome, maintaining a healthy gut condition and digestive health [136,137]. Research on both humans and animals indicates that synbiotics are capable of facilitating anthropometric features, including weight reduction in individuals with obesity [18,110], as well as aiding in the weight increase in children suffering from malnutrition [111,112,113]. These findings highlight the dual benefits of synbiotics in managing weight-related issues. This dual capability emphasises the promise of synbiotics in addressing a range of health issues associated with nutrition and weight [138]. However, it can be noted that most of these studies only reported the role of synbiotics on anthropometric outcomes, but the results regarding the role of synbiotics in malnutritional-related biomarkers such as gut microbiome, neuroactive compounds, and appetite hormones are very limited, and further research is warranted. It can be fully understood the impact of synbiotics on malnutrition, as well as how synbiotics affect individuals with malnutrition at both the physiological and pathological levels.

A study by Nuzhat et al. [111] found that in children with severe acute malnutrition, the rate of weight gain was higher in those supplemented with probiotics (Bifidobacterium. infantis EVC001) alone compared to those who were administered synbiotics (Bifidobacterium. infantis EVC001 with prebiotic Lacto- N-neotetraose) supplementation. Although synbiotics integrate probiotics and prebiotics, and theoretically their synergistic action can enhance the survival and activity compared to single biological functions, in some cases, the prebiotics might not directly provide adequate support for the gut microbiota. However, probiotics are live microorganism that work directly within the gut to improve the balance of the bacterial community. They can rapidly replenish beneficial bacteria in the gut, promoting digestion and absorption, thereby effectively enhancing weight gain in malnourished children [111]. It is hoped that longer-term supplementation is required to understand the sustained effects of probiotics and synbiotics on the outcomes in malnourished children [111].

### 3.4. Postbiotics

Over the past few years, a novel area of research has emerged in which probiotic derivatives evaluate the origin composition, mode of action, and potential advantages. This development has led to the emergence of globally accepted terms such as postbiotics. [139]. According to the International Scientific Society for Probiotics and Prebiotics (ISAPP), postbiotics are defined as “a preparations of non-viable microorganisms and ingredients that provide health benefits to the host.” [139].

ISAPP also emphasises that postbiotics must include cell components or microbial cells, which are attenuated without or with metabolites and have been proven to have a positive impact on health [139]. It is essential to characterise the microbial composition of the preparation before attenuation in order to consider it as a postbiotic. It has been suggested that postbiotics can be defined as microbial factors derived from foods fermented by identified microorganisms, as opposed to conventional foods fermented by unidentified microbial cultures. Therefore, postbiotics usually refer to secreted metabolites such as cell-free supernatants, cell-free extracts, peptides and proteins, enzymes, and SCFAs [140].

#### Effect of Postbiotics on Appetite Regulation

The efficacy and health benefits of 24 weeks supplementation with inulin-propionate ester as postbiotic was investigated in a primary cultured human colonic cell model that mimicked of overweight status, which revealed that inulin-propionate ester significantly reduced body weight. In addition, acute ingestion of 10 g inulin-propionate ester notably boosted postprandial plasma levels of GLP-1 and PYY and decreased energy intake. The inulin-propionate ester treatment led to a significant decrease in subjective appetite ratings following meals. A notable trend was observed, indicating an 8.7% (73 kcal) reduction in food intake, which implies that propionate might affect appetite and energy intake via pathways independent of GLP-1 or PYY release [114].

In another study on the effects of the postbiotic EPS-layer protein from *Leuconostoc mesenteroides* DH 1606 (LCM6) and *L. mesenteroides* DH 1608 (LCM8) supplementation on high fat diet mice, postbiotics significantly improved the dysbiosis of the gut microbiota induced by a high-fat diet, increasing the abundance of beneficial bacteria such as bifidobacteria and lactic acid bacteria, while reducing the levels of harmful bacteria. Moreover, no adverse effects associated with the postbiotic supplementation were observed during the intervention study. This indicates the application of postbiotics in obesity intervention, with the potential to improve obesity and related metabolic disorders induced by high-fat diet by regulating intestinal microbiota and metabolic pathways [141]. However, the current understanding of how postbiotics affect anorexia nervosa (AN) and malnutrition is still limited. Further research is required to investigate the optimal dosage, mechanism of action, and treatment protocols for addressing undernutrition [142].

### 3.5. Paraprobiotics

Paraprobiotics have also been employed to define an inactivated, non-viable microorganism, whether ruptured or intact, and confers health benefits [143]. Specifically, the term “paraprobiotic” denotes bacteria that have been rendered inactive. Compared to traditional probiotics, paraprobiotics are prepared by different inactivation methods such as heat treatment, high voltage, supercritical carbon dioxide technology, pulsed electric fields, etc., which affect their cellular components such as DNA or proteins. These methods are able to retain their cellular components and some of their biological activity while avoiding the safety risks that can be posed by live bacteria [144]. The research on the direct effects of postbiotics on appetite regulation is still limited.

#### Effect of Paraprobiotics on Neuroactive Compounds Regulation

Paraprobiotics can indirectly influence appetite through the production of SCFAs; SCFAs stimulate the secretion of gut hormones like PYY and GLP-1, which enhance satiety and reduce food intake [145]. In most studies on the intervention of obesity by paraprobiotics, the primary focus has been on changes in body measurements, such as significant weight loss. However, few studies have reported changes in indicators related to the gut–brain axis. However, there is a potential link between body weight changes and the gut–brain axis. The first pathway is the immune and inflammatory pathway. Obesity is often accompanied by chronic low-grade inflammation, which can be alleviated by modulating the gut microbiota. Paraprobiotics may indirectly affect weight and metabolic health by improving gut barrier function and reducing the production of inflammatory mediators. This immune regulatory effect is also closely related to the function of the GBA [146]. The second pathway is the interaction between the microbiota and host metabolism. Studies have shown that the composition of the gut microbiota is closely related to the state of obesity. For example, the relative abundance of Firmicutes is higher, while that of Bacteroidetes is lower in the gut microbiota of obese individuals. By modulating the gut microbiota abundance, paraprobiotics may improve this imbalance, which in turn affects energy metabolism and body weight [147]. Some studies have shown that paraprobiotics can affect the action of neurotransmitters by regulating the expression of neurotransmitter receptor genes. For example, one study found that heat-inactivated Enterococcus faecalis EC-12 supplementation altered the expression of neurotransmitter receptor genes in the prefrontal cortex of mice and attenuated anxiety-like behaviour [148].

Although it is widely accepted that nutraceuticals including probiotics, prebiotics, synbiotics, postbiotics, and paraprobiotics have the potential to enhance gut health by modulating gut microbiota composition and improving GBA-related biomarkers, this perspective is oversimplistic and ignores the complexity of individual responses. The assumption that these interventions are generally effective for addressing malnutrition is problematic because it does not account for the significant variability in gut microbiota composition and GBA-related biomarkers among individuals with different types of malnutrition. In reality, the efficacy of nutraceutical interventions is highly dependent on individual genetic and microbiome profiles, which can vary widely even within the same category of malnutrition. This variability implies that a generic approach to nutraceutical supplementation is unlikely to produce consistent or optimal results. Instead, precision nutrition that takes into account the specific nutritional needs and gut microbiome characteristics of an individuals is essential for choosing the most appropriate nutraceuticals. However, even precision nutrition faces challenges of accurately categorising and addressing various types of malnutrition, highlighting the need for more nuanced and personalised strategies.

## 4. Challenges and Future Prospectives

The rise of malnutrition poses a major global health challenge. Traditional interventions often fail to address individual differences. However, personalised nutraceutical strategies offer dietary guidance that is specifically tailored to meet the unique nutritional requirements of each individual. While there is no universally accepted definition of personalised nutrition, recommendations are generally formulated based on an individual’s behaviours, biological characteristics, and the interaction between these factors [149]. The goal of personalised nutraceuticals is to improve dietary habits to prevent or manage chronic diseases, thereby promoting better public health outcomes [150].

### 4.1. Approaches in Nutraceutical Interventions for Individuals with Malnutrition

The development of a prediction framework for individuals with malnutrition responses to nutraceutical interventions will depend on iterative methods that can span the entire axis of translation, incorporating feedback loops between computational models of host–gut microbiome interactions [151]. Currently, there are in vitro and in vivo approaches available for the rational design and testing of personalised nutraceutical interventions for individuals with malnutrition, as shown in Table 4. Typically, in vitro approaches for testing these nutraceutical predictions examine how nutraceuticals may impact on the microbial community and biomarkers related to the GBA, and ultimately in vivo studies are conducted to assess the effect of these nutraceutical interventions on host health and physiology in both humans and animals.

#### 4.1.1. In Vitro Approach

Creating a predictive framework based on how nutraceuticals influence host health requires quantitative and dynamic comprehension of the characteristics anticipated to be predicted within the system. However, obtaining time-resolved and continuous data from in vivo studies is often challenging. Therefore, in vitro approaches offer a balance between flexibility and accuracy. In an in vitro model, microbial community composition and environmental variables can be precisely regulated, and the sampling time scale can be tailored as required [155]. This makes in vitro models highly suitable for dissecting variable responses to nutraceutical interventions and testing specific hypotheses related to host-nutraceutical–microbial interactions [156]. For example, in vitro continuous culturing methods can be a useful tool for determining how microbial community and biomarkers related to GBA changes in the presence of nutraceuticals with the physiologically relevant conditions. This method was developed by Macfarlane et al., designed to reflect different large intestine niches along the GIT. It has been demonstrated to replicate the bifidogenic effects of GOS seen in a human trial and offers a comprehensive prediction of where specific fermentation processes might occur in the gut [157,158]. In vitro approaches have been crucial in identifying which nutraceutical interventions are appropriate for animal models and human trials. However, they are constrained by the fact that many human symbiotic microorganisms are challenging to cultivate. The complexity of the human digestive system challenges experimental replication and affects the accuracy of microbial community dynamics studies. The absence of multiple tissue interactions and metabolites accumulation creates an artificial gut environment. This results in the “bottle effect”, in which the ecological characteristics of the microbial community in vitro start to differ from the actual situation in vivo [159]. However, advancements in culture techniques, tissue cultures, and the establishment of diverse gut microbiota libraries, such as the Global Microbiome Conservancy, have significantly enhanced the scope of in vitro research. Moreover, studies integrating host tissues also improve the relevance of these systems by incorporating diverse human-derived strains and host–microbe crosstalk [160].

#### 4.1.2. In Vivo Approach

To predict personalised responses to nutraceutical interventions in vivo, it is essential to have sufficient variability within the study population, collection of relevant biological measures that act as covariates, and robust methods for characterising the response or results. In preclinical animal models or human populations, capturing phenotypic and genetic through sampling is crucial for developing personalised nutraceutical prediction. In nonhuman trials, the colonisation of human faecal communities using germ-free mice and mice treated with an antibiotic mixture further expands the scope of examination for differences in response to microbial-driven nutraceutical inputs [161]. Non-human animal models offer detailed spatiotemporal insights into immune function, digestion, and physiology after nutraceutical interventions [162]. Key considerations for the rigor of these experiments and their applicability to humans include addressing concerns related to exposure–dose correlation, appropriate use of controls, and potential biases arising from handling [163].

In human trails, nutraceutical intervention research has spanned from observational studies in prospective cohorts to randomised controlled trials (RCTs). Among these different types of approaches, the selection of study design, the length of the intervention period, and the sampling timeline are essential for elucidating the relevant biological mechanisms and timelines associated with the changes induced by nutraceutical interventions [164]. The duration of nutraceutical interventions targeting gut microbiota and other biomarkers outcomes may vary. Longer interventions make trials impractical [165]. When particular microbes and pathways are already established in the system, some certain areas of gut microbiota activity can rapidly adapt to short-term nutraceutical interventions [165]. The immediate interactions between nutraceutical and gut microbiota can be detected in the metabolome or physiological biomarkers shortly within hours of intake. However, to achieve considerable alterations in the functional output and the composition of the gut microbial community, long-term nutraceutical interventions may be necessary [166].

### 4.2. Vision for the Future: Multi-Omics Technologies Driven Tailored Nutraceuticals in Malnutrition

It is has become evident that the gut microbiota significantly influences how individuals respond to nutraceutical interventions. By incorporating microbiome data along with clinical data and responses to standardised nutraceuticals into predictive trials, and these studies can be used to develop personalised interventions that are more effective than current standard practices [167]. However, there are potential shortcomings, as many existing commercial precision dietary and nutraceutical interventions are still in their early stages and often overestimated in loosely regulated markets. In these stages, more evidence needs to be gathered from hypothesis-generating, well designed human observational studies, particularly those that combine intensive clinical, phenotypic, and behavioural information with gut microbiome analysis [167,168]. Statistical learning models that predict personalised phenotypic responses to nutraceuticals interventions can be constructed using large, diverse, and densely phenotype human populations [169]. However, these methods are limited because they may not offer detailed mechanistic insights and depend on training cohorts. Predictions may deteriorate when new data are derived from individuals significantly different from the training cohort. To address this, it is necessary to utilise the expanding knowledge base to develop superior mechanistic models grounded in causally validated microbe–microbe and host–microbiome interactions, thereby enhancing personalised predictions. Advances in host–microbiota metabolism modelling now allow for predictions of responses to high-throughput, personalised nutraceutical interventions [170]. To validate these computational predictions, in vitro models must be controlled for high-resolution, spatially resolved, and longitudinal sampling. For instance, data from in vitro batch or continuous culture systems can directly validate metabolic model predictions of personalised SCFA production tailored to specific nutraceuticals [155]. Beyond validating existing models, complex in vivo non-human animal models and in vitro models can explore uncharacterised host-microbiome relationships [160,162]. These experimental insights can enrich knowledge base and support improved mechanistic modelling. Ultimately, human observational and intervention studies are essential for validating personalised dietary and nutraceutical interventions and obtaining regulatory approval. Quantitative validation in both in vivo and in vitro models is essential for optimising human health, nutraceuticals, and overall well-being through gut microbiota modulation, and further indicate GBA related biomarkers, as shown in Figure 3.

The future of nutrition hinges on integrating multi-omics technologies to develop tailored nutraceuticals that target the underlying causes of malnutrition, which aims to reveal all biological molecules involved in the structure, function, and dynamics of a cell, organism, or all organisms in a particular environment [171]. These include a comprehensive analysis of genes (genomics), microbiota (microbiomics), metabolites (metabolomics), and food (foodomics) [172]. There is an extremely important link between bioactive food components and cellular processes, and this link shows significant differences at different molecular levels. Specifically, these differences are not only reflected in the nutrigenomics that affects messenger RNA, but also in the nutrigenetics that affects DNA [173]. Given that a single omics technique fails to capture the full scope of malnutrition, the application of multi-omics techniques has been widely recognised to provide a more comprehensive perspective [172]. Advances in multi-omics have elucidated how nutrients influence individuals at the molecular and cellular levels, thereby enabling personalised nutraceutical interventions to consider a broader range of factors, including physiology, foodomics, genomics, and microbiome [173]. However, applying these technologies on an individual basis is time-intensive and expensive. Metabolic typing, which categorises individuals into subgroups based on metabolic phenotypes, such as biochemical, gut microbiome, and metabolomics data, emerges as a more viable and cost-efficient alternative for precision nutraceutical strategies [174].

#### 4.2.1. Genomics of Malnutrition

Exploring the human genome has been a path to deeper insights into the variations that exist between individuals. Genome-wide association studies (GWAS) have transformed the field of multifactorial disease genetics, with the goal of pinpointing genomic variants to elucidate the genotype–phenotypic connections associated with disease susceptibility [175]. Despite the ongoing shortfall in studies featuring large sample sizes with ethnic diversity, GWAS has effectively uncovered novel genetic variants and mechanisms underlying conditions like nutritional related disorders, including the identification of genes such as FTO (fat mass and obesity-associated protein), leptin receptor, leptin, TNF-α, and interleukin, that are linked to metabolic issues in malnutrition [175]. Notably, among numerous metabolism-related genes, FTO was identified as the strongest predictor of polygenic overnutrition. The FTO gene is recognised for its role in regulating eating behaviour and appetite, is associated with a higher risk of overnutrition primarily due to increased food intake and preference for high-energy and high-fat foods [176]. As research accumulates and technology advances, the pace of identifying metabolic-related candidate genes is accelerating. Considering the vast and intricate nature of genomic data, sophisticated analytical tools such as machine learning algorithms and deep learning techniques are essential for identifying hidden genomic connections, formulating hypotheses and innovative models, and generating predictions. Machine learning algorithms are particularly well-suited to data-intensive fields such as genomics, as they are engineered to autonomously identify patterns within complex datasets [177]. Recognising the vital importance of genetic information in precision nutrition, scientists have incorporated specific genes and their associated single nucleotide polymorphisms (SNPs) into tailored nutraceuticals interventions. The use of SNPs has emerged as a key approach in personalised nutraceuticals interventions. As more metabolic-related genetic variants are identified, genetic data are anticipated to help pinpoint populations with a higher susceptibility to metabolic-related disorders. In cases where a genetic predisposition exists, there is potential to intervene earlier to prevent metabolic malnutrition diseases [178].

#### 4.2.2. Harnessing Microbiomics for Nutraceutical Interventions

The emerging field of microbiomics holds great promise for developing personalised nutraceutical interventions to address malnutrition. By understanding the complex interactions between the gut microbiome, host health and nutraceutical interventions, researchers can design targeted therapies that modulate gut microbial communities to improve metabolic outcomes [179]. Diet significantly influences the gut microbiota composition and its interaction with nutraceuticals such as prebiotics and probiotics is a well-known health promotion strategy. Overnutrition typically results from a sustained positive energy balance caused by unbalanced energy intake and energy expenditure, and traditional weight loss approaches primarily focus on increased physical activity and low-calorie diet. However, the impact of individual genetic factors and microbiota-related mechanisms on energy metabolism is often overlooked. Research indicates that gut microbiota profiles may elucidate variations in weight loss among individuals undergoing the same nutraceutical intervention [180]. In particular, dysbiosis of the gut microbiota leads to the depletion of bacteria that produce SCFAs, which have been implicated in the aetiology of malnutrition by inducing inflammation [181]. Additionally, a mixed of SCFAs supplementation on the rectal increased fatty acid oxidation, plasma PYY levels and energy expenditure in overweight and obese individuals [182]. It can be seen that precision nutraceutical can only fulfil its potential by paying closer attention to the intricacies of interactions between nutraceutical and microorganisms [183]. A study demonstrated that diet and the gut microbiome are more influential than genetics in explaining inter-individual variability in metabolism, based on an assessment of plasma metabolites. They highlighted that nutraceutical quality predicted by machine learning models using an individual’s plasma metabolome is significantly correlated with nutraceutical quality assessed by food frequency questionnaires [184]. This finding underscores the potential of omics technologies to use the gut microbiome as a predictive indicator of individual responses to the same nutraceutical intervention. Consequently, identifying microbiomics involved in altering human gut homeostasis through novel sequencing approaches could pave the way for the development of tailored nutraceutical treatments [185].

While clinical trials offer some insights for nutraceutical research, the data available in the field of microbiomics remain limited. In terms of the microbiome data utilised, most studies depend on 16S rRNA gene sequencing for microbial analysis. Although metagenomic data derived from whole-genome sequencing are becoming more accessible, their application in precision nutraceutical trials is not widely used [185]. For example, there is still inadequate evidence to regard baseline gut microbiota as a dependable predictor of weight gain or loss associated with malnutrition. A significant challenge is the lack of consistency among studies examining the role of microbiota in the pathophysiology of undernutrition and overnutrition, which hampers the ability to establish a causal link between body weight changes and gut microbiota composition. Given the complexity of identifying or selecting the appropriate variants, there is a need for variant calling algorithms based on large sample cohorts to facilitate overnutrition or undernutrition gene mapping [186].

#### 4.2.3. Metabolomics and Malnutrition—Unveiling the Metabolic Fingerprint

Metabolomics seeks to characterise the metabolome, which serves as a chemical manifestation of a biological phenotype. Commonly utilised biological samples in metabolomics research include blood (e.g., plasma, serum and whole blood), saliva, faecal and urine [187]. Metabolomics offers significant potential for the development of personalised nutraceutical interventions for malnutrition. By capturing a comprehensive snapshot of an individual’s metabolic profile, metabolomics can reveal unique biochemical profiles that reflect their metabolic health, nutritional status and susceptibility to related conditions [188]. A precision nutraceutical approach requires a deep understanding of how interactions between genetics, metabotypes, and diet influence the levels of nutraceutical biomarkers. SNPs as genetic variants can influence metabolic differences and determine an individual’s nutraceutical needs and responses to different diets. therefore, biomarkers of food intake could be highly valuable in precision nutrition interventions by offering precise and objective assessments of nutrient intake and dietary habits. By identifying specific biomarkers of food intake, it is possible to predict individual responses to nutraceutical interventions, thereby increasing the effectiveness of precision nutraceutical interventions [189]. However, the application of biomarkers of food intake in precision nutraceutical research is still restricted, indicating a need for more in-depth incorporation and investigation into future studies, this highlights the difficulties associated with integrating biomarkers of food intake into multi-omics technologies [190]. While metabolomics has demonstrated promise in guiding precision nutraceutical assessments, its advantages over traditional biochemical markers are still a matter of debate. The most critical challenge lies in identifying which genetic variants or potential biomarkers of food intake should be prioritised for further study [191]. While biomarkers of food intake offer more objective insights compared to conventional nutritional assessment methods, metabolomics faces several challenges. These include higher invasiveness, inability to reflect long-term dietary patterns, lack of sensitivity or specificity for certain foods or nutraceutical, most critically, valid biomarkers are still lacking for many foods and nutrients [192]. Therefore, it is essential to focus on identifying new biomarkers for specific nutraceutical intake and rigorously validating these candidates against these criteria, including time-response, dose-response, reproducibility, reliability, and stability. This will help overcome current barriers to their use in precision nutraceutical intervention studies [193].

It can be seen that metabolomics is a powerful tool in the field of personalised nutrition, which providing detailed insights into the metabolic fingerprint of individuals that can guide the development of tailored nutraceutical interventions [194]. By identifying specific metabolic pathways affected by malnutrition, metabolomics aids in understanding how nutraceuticals affect metabolic well-being and provide a scientific basis for personalised dietary recommendations. Nutraceuticals include a range of bioactive compounds such as probiotics and prebiotics that have demonstrated their potential in the fight against malnutrition by modulating cellular signalling pathways, reducing oxidative stress and enhancing gut health to improve overall nutritional status [10]. The integration of multi-omics technologies including genomics, microbiomics and metabolomics, presents a future vision where personalised nutraceutical interventions are driven by a comprehensive understanding of an individual’s genetic makeup, gut microbiota composition and metabolic profile. This method has the potential to transform malnutrition management by offering more targeted and effective treatment strategies. Future research directions should concentrate on deciphering the mechanisms by which nutraceuticals interact with the gut microbiome to shape metabolic health and on developing biomarkers that can predict individual reactions to particular nutraceutical treatments [10]. Additionally, it is imperative to investigate the long-term effects and sustainability of personalised nutraceutical interventions in improving GBA health and combating malnutrition [195].

## 5. Conclusions

The exploration of nutraceuticals as a novel approach to modulate the GBA and combat malnutrition represents a major advance in the field of personalised nutrition and wellness. As highlighted in this review, nutraceuticals such as prebiotics, probiotics, and synbiotics have shown potential benefits in restoring gut microbiota composition, improving GBA related biomarkers, and addressing malnutrition. The intricate interplay between the gut microbiota and the brain underscores the importance of targeted interventions that leverage these bioactive compounds to optimise health outcomes. However, despite promising developments, several challenges remain. Firstly, the variability of individual microbiomes and the complexity of microbial metabolites require further research to elucidate the precise mechanisms underlying nutraceutical impacts on the GBA. Secondly, future studies aimed at reinforcing the evidence for personalised nutraceutical interventions should encompass large-scale RCTs with longer intervention periods, with a focus on evaluating the long-term enhancements of individualised nutrition on diverse health outcomes. In conclusion, nutraceuticals hold immense potential for revolutionising the management of malnutrition and promoting GBA health. As our understanding of the GBA and the role of nutraceuticals continues to evolve, the integration of personalised nutrition approaches has the potential to completely transform the way we prevent and treat malnutrition.

## Figures and Tables

**Figure 1 nutrients-17-01551-f001:**
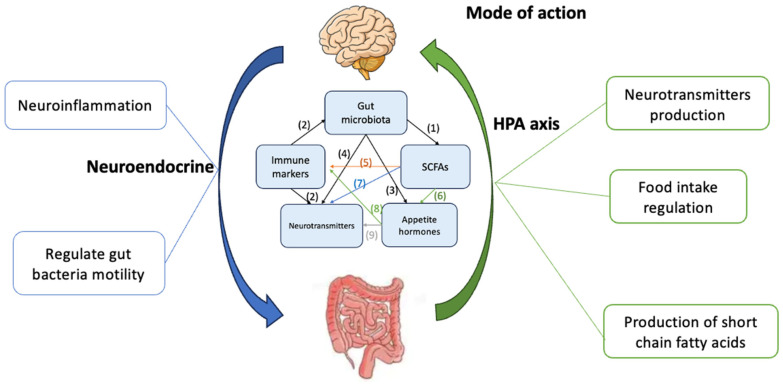
Communication routes of GBA biomarkers in individuals with malnutrition. There are five potential biomarkers linked with malnutritional status including gut microbiota, SCFAs, immune markers, appetite hormones, and NTs. Nine routes of communication are: (1) Gut microbiota effects on SCFAs regulation; (2) Immune markers effect on gut microbiota and NTs regulation; (3) Gut microbiota effects on appetite hormones regulation; (4) Gut microbiota effects on NTs regulation; (5) SCFAs effect on immune markers regulation; (6) SCFAs effect on appetite hormones regulation; (7) SCFAs effect on NTs regulation; (8) Immune markers effect on appetite hormones regulation; (9) NTs effect on appetite hormones regulation.

**Figure 2 nutrients-17-01551-f002:**
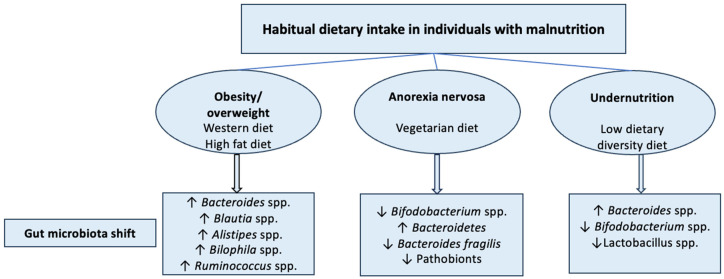
Summary of alternations in the gut microbiota associated with habitual dietary intake in individuals with malnutrition. (↓ decrease, ↑ increase).

**Figure 3 nutrients-17-01551-f003:**
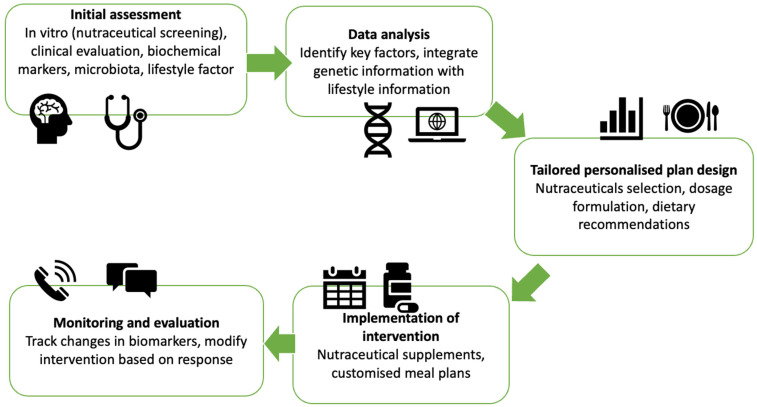
Process of a tailored nutraceutical intervention. This figure illustrates the process of a tailored nutraceutical intervention, which is a personalised approach to improve health outcomes. The process is divided into five main steps including Initial Assessment, Data Analysis, Tailored Personalised Plan Design, Implementation of Intervention and Monitoring and Evaluation, designed to optimise health outcomes through personalised nutraceutical interventions. Each step is crucial for ensuring that the intervention is both effective and tailored to the unique needs of the individual.

**Table 1 nutrients-17-01551-t001:** Individuals with malnutrition and their characteristics.

	Energy Imbalance	Manifestation	Characteristic	Examples	Ref.
Overweight/ obesity	Positive	Excessive nutrients intake, accumulation of excess body fat	BMI ≥ 25 for overweight, ≥30 for obesity	Obesity, metabolic syndrome	[5,6]
Anorexia nervosa	Negative	Severe restriction of food intake	BMI < 18.5, severe restriction of food intake, extreme weight loss	Anorexia nervosa	[7,8]
Undernutrition	Negative	Insufficient intake or absorption of essential nutrients	BMI < 18.5, low body weight, stunted growth, reduced MUAC, weakened immune system	Kwashiorkor, severe acute malnutrition, failure to thrive	[5]

Abbreviations—BMI: body mass index, MUAC: mid-upper arm circumference.

**Table 2 nutrients-17-01551-t002:** Summary of changes in the malnutritional status associated with GBA and potential biomarkers linked with such changes.

GBA Biomarkers	Obesity	Anorexia Nervosa	Undernutrition	Physiological Relevance	Ref.
SCFAs	↓ Total SCFAs	↓ Butyrate; ↓ propionate	↓ Butyrate; ↓ propionate	Energy source, gut barrier integrity, anti-inflammatory effects, appetite regulation, gut microbiota metabolism	[86,87]
Neurotransmitters	↓ Serotonin	↓ Serotonin; ↓ dopamine	↓ Serotonin; ↓ dopamine	Influence mood, appetite, energy expenditure, and reward pathways	[85,88,89]
Appetite hormones	↓ PYY; ↓ GLP-1; ↑ leptin	↓ Leptin; ↑ ghrelin; ↑ PYY	↑ ghrelin; ↓ leptin	Appetite and body weight regulation, energy homeostasis regulation	[90,91]
Immune markers	↑ TNF-α; ↑ IL-6; ↑ CRP; ↓ IL-10	↑ TNF-α; ↑ IL-1β	↑ TNF-α	Metabolic regulation; Increased gut permeability indued by pro-inflammatory markers	[88,92]

Abbreviations—Increase (↑); Decrease (↓); SCFAs: short-chain fatty acids; PYY: peptide YY; GLP-1: glucagon-like peptide; TNF-α: tumour necrosis factor-a; IL: interleukin; CRP: C-reactive protein; IL-1β: interleukin-1 beta.

**Table 3 nutrients-17-01551-t003:** The effects of nutraceuticals on individuals with malnutrition.

Nutraceuticals	Study Population	Intervention Time and Daily Dose	Model	Main NDs Related Findings	Ref
Probiotics studies					
Probiotic: *Bifidobacterium pseudocatenulatum* CECT 7765	Obesity	14 weeks, 1 × 10^9^ CFU/day	Mice	↓ Weight, ↑ leptin receptor mRNA, ↓ Leptin, ↓ DA, ↓ NE, ↑ 5-HT concentrations in the hypothalamus	[12]
Probiotic 1: *Lactobacillus sakei* OK67, probiotic 2: *Lactobacillus sakei* PK16	Obesity	4 weeks; 2 × 10^9^ CFU/day	Mice	In both treatments: ↓ Firmicutes, ↓ Proteobacteria, ↑ Verrucomicrobia, ↓ delta-Proteobacteria, ↓ Deferribacteres, ↓ weight; ↓ TNF-α; ↓ NF-κB; ↓ anxiety like behaviours	[13]
Probiotic: *Lactobacillus paracasei* HII01	Obesity	12 weeks, 1 × 10^8^ CFU/day	Rats	↓ Weight; ↓ ratio of F/B; ↓ IL-1 mRNA; ↓ IL-6 mRNA	[96]
Probiotic 1: *Lactobacillus brevis* DPC6108 probiotic 2: *Lactobacillus brevis* DSM32386	Obesity	12 weeks, 1 × 10^10^ CFU/day	Mice	In both treatments: ↓ weight, ↑ faecal flora diversity, ↑ ratio of F/B, ↑ GABA in the small intestine	[97]
Mixed probiotic supplementation: *Lactobacillus salivarius* CUL61, *Lactobacillus paracasei* CUL08, *Bifidobacterium bifidum* CUL20, and *Bifidobacterium animalis* subsp. *lactis* CUL34	Obesity	12 weeks, 5 × 10^8^ CFU/day	Mice	↓ Weight, ↓ Lactobacilli; ↓ Enterobacteria, ↓ Coliforms↓ Yeast, ↑ Enterococci, ↑ IL-10 mRNA, ↓ IL-18 mRNA	[98]
Probiotic: *Lactobacillus reuteri* MM4–1A	Obesity	6 weeks, 5 × 10^9^ CFU/day	Mice	↓ Ratio of F/B, ↓ weight, ↑ TNF-α, ↓ IL-1b, ↓ IL-6 in the hippocampus	[99]
Probiotics: Yoghurt containing *Lactobacillus delbrueckii* subsp. *bulgaricus* and *Streptococcus.thermophilus*	Patients with AN	10 weeks, 375 g yoghurt/day	Human	↑ Interferon-γ, ↑ CD4+/CD8+ ratio, ↑ T lymphocyte subset	[45]
Multi-probiotic supplementation: *Lactobacillus acidophilus*, *Bifidobacterium bifidum*, *Bifidobacterium lactis, Bifidobacterium longum, Lactobacillus rhamnosus*, *Lactobacillus reuteri*	Obesity and food addiction	12 weeks, each strain is 1.8 × 10^9^ CFU including: *Lactobacillus acidophilus, Bifidobacterium bifidum, Bifidobacterium lactis, Bifidobacterium longum.* The 1 × 10^9^ CFU/capsule including *Lactobacillus rhamnosus, Lactobacillus reuteri*	Human	↓ Weight, ↓ leptin, ↓ neuropeptide	[46]
Probiotic: *Bacteroides uniformis* CECT 7771	Food addiction	Rats that fasted 12 h and received a daily dose of 1 × 10^8^ CFU	Rats	The effects of *Bacteroides. uniformis* on the brain reward response are mediated by changes in the levels of DA, NE and 5-HT in the nucleus accumbens as well as in the expression of dopamine receptors in the prefrontal cortex and intestine. An increase in the OTUs and the phylogenetic diversity	[100]
Probiotic: yoghurt containing *Lactobacillus. Bulgaris*, *Streptococcus. thermophilus*	Two different situations: (1) Malnourished children; (2) Patients with AN	10 weeks, 125 g yoghurt/day	Human	In both groups: ↑ Interferon-γ	[101]
Probiotic: *Lactobacillus reuteri* DSM17938	Patients with AN	13 weeks, 2 × 10^8^ CFU/day	Human	↑ Weight, ↑ body mass index	[102]
Mixed probiotics supplementation: *Bifidobacterium breve*, *Bifidobacterium longum*, *Bifidobacterium infantis*, *Streptococcus thermophilus*, *Lactobacillus acidophilus*, *Lactobacillus plantarum*, *Lactobacillus paracasei*, *and Lactobacillus delbrueckii subsp. bulgaricus*	Translational activity-based anorexia	2 days, 1 × 10^9^ CFU/mL	Rats	An increase formation of GALT provided with probiotics supplementation, possibly related to gut microbiome, also contributes to the imbalanced levels of pro-inflammatory and anti-inflammatory cytokines observed in patients with AN.	[103]
Mixed probiotics supplementation: *Lactobacillus acidophilus*, *Bifidobacterium longum*, and *Enterococcus faecalis*	Participants on a high fat diet	4 months, 2 g probiotic powder/day (1.0 × 10^7^ CFU/g)	Human	↑ *Ruminococcaceae* and *Lachnospiraceae* family, ↓ *Bacteroidaceae* family	[104]
Mixed of probiotics strains in fermented milk: *Lactobacillus acidophilus* CUL60, *Lactobacillus acidophilus* CUL21, *Lactobacillus acidophilus* NCFM, *Bifidobacterialactis* HNO19, *Bifidobacteriaanimalis-supsplactis* CUL34, and *Bifidobacteriabifidum* CUL20	Obesity	3 months, 100 g/day (One fermented milk cup contained 10 × 10^9^ CFU)	Human	↓ Weight, ↓ leptin, ↓ SCFA, ↑ Lactobacillus, ↑ Bifidobacteria, ↑ Bacteroidetes, ↓ Firmicutes, ↓ ratio of F/B	[105]
Prebiotics studies					
Prebiotic-supplemented diet containing OFS	Overweight	13 days (1) 10 g OFS/day or (2) 16 g OFS/day	Human	In both treatments: ↑ PYY, ↑ GLP-1, ↓ energy intake. PYY and GLP-1 levels were significantly lower with 16 g/d OFS compared with 10 g/d OFS. Energy intake was significantly lower with 16 g/d OFS compared with 10 g/d OFS	[14]
Prebiotic-supplemented diet: OFS	Overweight and obese adults	12 weeks, 21g/day	Human	↓ Body weight, ↓ fat mass, ↓ energy intake, ↓ ghrelin	[15]
Prebiotic: OFS-enriched inulin	Overweight or obesity	16 weeks, 8 g/day	Human	↓ Weight; ↓ IL-6; ↑ *Bifidobacterium* spp; ↓ *Bacteroides vulgatus*	[106]
Prebiotic-supplemented diet: chicory-derived fructan	Healthy non-obese adults	2 weeks, 16 g chicory-derived fructan/day	Human	↑ PYY, ↑ GLP-1, ↓ hunger	[107]
Prebiotic: Inulin	Wild type mice	14 weeks, 7.5% inulin/day	Mice	PYY was reduced by 87%	[108]
Probiotic: *Saccharomyces. Boulardii* Prebiotic: FOS	Mimic of AN gut condition based on AN patients’ dietary pattern	16 days; *Saccharomyces. Boulardii*: 5 × 10^8^ CFU/day; FOS: 1.67 g/day	In vitro gut model system	In *Saccharomyces Boulardii* treatment: ↑ GABA and 5-HT in proximal, ↑ total bacteria in transverse colon. In FOS treatment: ↑ acetate, *Bifidobacterium* spp., *Roseburia* genus and total bacteria in proximal, transverse and distal colon; ↑ butyrate in proximal and distal colon; ↑ propionate, EPI and DA in proximal colon.	[109]
Prebiotic treatment: OFS Probiotics treatment: *Bifidobacterium animalis* subsp. *lactis*, synbiotic treatment: probiotic (*Bifidobacterium animalis* subsp. *lactis*) with prebiotic (OFS)	Rats with high fat diet-induced obese	8 weeks, prebiotic: 10% (*wt*/*wt*) OFS/day, probiotic: 1 × 10^10^ CFU/day, symbiotic: 10% (*wt*/*wt*) OFS with *Bifidobacterium animalis* subsp. *lactis* of 1 × 10^10^ CFU/day	Rats	In OFS treatment: ↑ GLP-1, ↑ PYY, ↓ leptin, ↑ *Bacteroides* spp., ↑ *Lactobacillus* spp., ↑ *Bifidobacterium* spp., ↑ *Bifidobacterium. animalis,* ↓ *C. coccoides,* ↓ *C. leptum,* ↓ *Clostridium* Cluster XI and I, ↓ *Enterobacteriaceae,* ↓ the ratio of F/B. In *Bifidobacterium animalis* subsp. *lactis* treatment: ↑ GLP-2, ↑ *Bifidobacterium.animalis*	[16]
Synbiotics studies					
Synbiotic treatment: probiotic (*Bifidobacterium animalis* subsp. *lactis*) with prebiotic (polydextrose), probiotic treatment: *Bifidobacterium animalis* subsp. *lactis*	Overweight and obese	6 months, synbiotics: 12 g/day of polydextrose and 10^10^ CFU of *Bifidobacterium animalis subsp.* lactis *plus,* probiotic: 10^10^ CFU/day	Human	Synbiotics treatment: ↓ weight, ↑ *Akkermansia,* ↑ *Christensenellaceae,* ↑ *Methanobrevibacter,* ↓ *Paraprevotella* Probiotic treatment: ↑ *Lactobacillus,* ↑ *Akkermansia*	[110]
Synbiotic: probiotic (*Lactobacillus rhamnosus* CGMCC1.3724) with prebiotic (OFS and inulin)	Obese	24 weeks, 1.6 × 10^8^ CFU of *Lactobacillus rhamnosus* CGMCC1.3724 and 300 mg of a mix of OFS and inulin/day	Human	↓ Weight; ↓ leptin; ↑ *Lachnospiraceae*	[17]
Synbiotic: mixed probiotic (*Lactobacillus acidophilus*, *Bifidobacterium lactis*, *Bifidobacterium longum*, *Bifidobacterium bifidum*) with prebiotic (galactooligosaccharide)	Overweight	3 months, 15 × 10^9^ CFU of mixed strains (*Lactobacillus acidophilus* DDS-1, *Bifidobacterium lactis* UABla-12, *Bifidobacterium longum* UABl-14, and *Bifidobacterium bifidum* UABb-10) and 5.5 g galactooligosaccharide/day	Human	↑ *Bifidobacterium*; ↑ *Lactobacillus*; ↑ *Ruminococcus*; ↑ *Verrucomicrobiae*	[18]
Probiotic: *Bifidobacterium. infantis* EVC001), Synbiotic treatment: probiotic (*Bifidobacterium. infantis* EVC001) with prebiotic (Lacto-N-neotetraose [LNnT])	Children with severe acute malnutrition	4 weeks, probiotic: 8 × 10^9^ CFU/day; Synbiotic: probiotic (8 × 10^9^ CFU) plus 1.6 g prebiotic/day	Human	↑ Rate of weight gain in probiotic group compared to synbiotic group	[111]
Synbiotic: mixed probiotic (*Lactobacillus acidophilus*, Lactobacillus rhamnosus, *Lactobacillus bulgaricus*, *Lactobacillus casei*, *Bifidobacterium infantis*, *Bifidobacterium breve*, and *Streptococcus thermophilus)* with prebiotic (FOS)	Children with FTT	30 days, synbiotic: probiotic (1 × 10^9^ CFU) plus 1.0 g prebiotic/day	Human	↑ Weight	[112]
Synbiotic: probiotic (*Bacillus coagulans*) with prebiotic (FOS)	Children with FTT	6 months, 100 mg FOS and 150 million spore *Bacillus coagulans*/day	Human	↑ Weight, ↑ BMI	[113]
Postbiotics studies					
Inulin-propionate ester	Overweight (cultured human colonic cell model)	24 weeks, 10 g/day	Human	↓ Weight, ↑ PYY, ↑ GLP-1	[114]
Acetate sodium	Overweight/obese men	3 days, distal and proximal colon: (100 or 180 mmol/L dissolved in saline 120 mL)	Human	Distal colon: ↑ PYY, ↓ TNF-α; Proximal colon: no significant difference.	[115]
Paraprobiotics studies					
Heat-killed LP28	Overweight	12 weeks, 7.5 mL (10^11^ cells)	Human	↓Body fat mass, ↓ BMI, ↓ waist circumference, ↓ body fat percentages	[116]
Fragmented CP1563	Overweight and mildly obese	12 weeks, 200 mg paraprobiotics in a 500 mL beverage	Human	↓ Body fat percentage, ↓ whole body fat, ↓ visceral fat	[117]

Abbreviations—↓ decrease; ↑ increase; mRNA: messenger RNA; DA: dopamine; NE: norepinephrine; 5-HT: serotonin; F/B: Firmicutes/Bacteroidetes; IL: interleukin; TNF-α: tumour necrosis factor-α; NF-κB: Nuclear factor-κB; GALT: gut-associated lymphoid tissue; FOS: fructooligosaccharides; OTU: operational taxonomic units; CFU: colony-forming units; AN: anorexia nervosa; HFD: high fat diet; OFS: oligofructose; GLP: glucagon-like peptide; FTT: failure to thrive; GABA: Gamma-aminobutyric acid; SCFAs: short-chain fatty acids; PYY: peptide YY; BMI: body mass index.

**Table 4 nutrients-17-01551-t004:** In vitro and in vivo approaches to exploring how nutraceutical intervention impact on gut microbiota and biomarkers related to GBA in individuals with malnutrition.

Approach	Main Points	Advantages	Disadvantages	Ref.
In vitro	Studies conducted outside a living organism (e.g., cell cultures, gut organoids, or microbiome and colon simulations). Test bioavailability, absorption, and metabolism of nutraceuticals. Study direct effects on gut microbiota and epithelial cells.	No ethical concerns High-throughput screening	Limited relevance to whole organism physiology Cannot fully replicate gut brain axis interactions.	[109,152]
In vivo	Studies conducted within a living organism in animal models and human clinical trials. Administer nutraceuticals orally or through diet. Study systemic effects on gut microbiota and gut brain axis biomarkers.	Captures systemic and physiological effects on GBA interactions. More aligned with human biology and directly relevant to human outcomes	Ethical concerns Long term study is expensive and challenging	[153,154]

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
