# Peer review of "Nutraceuticals for Gut–Brain Axis Health: A Novel Approach to Combat Malnutrition and Future Personalised Nutraceutical Interventions"

_nutrients, 2025, doi:10.3390/nu17091551_

Round 1

Reviewer 1 Report

Comments and Suggestions for Authors

The authors of the manuscript "Nutraceuticals for Gut-Brain Axis Health: A Novel Approach to Combat Malnutrition and Future Personalized Nutraceutical Interventions"aim to explore the future of personalized nutraceutical interventions tailored to individual genetics and microbiome characteristics, providing a framework for the development of precision formulations to optimize health outcomes. The manuscript presents an interesting and relevant topic, and I recommend minor revisions to enhance its clarity and depth.

  1. The authors should establish a clearer link between short-chain fatty acids (SCFAs) and neuroinflammation in the section "Effects of gut microbiota on SCFA regulation."
  2. The discussion section would benefit from a more in-depth exploration of metabolomics, nutraceuticals, their applications in personalized nutrition, and potential future research directions in the field.

Author Response

Dear reviewer, thank you so much for your feedback and that is really helpful for us. We have corrected the manuscript based on your comments. 

Comments 1. The authors should establish a clearer link between short-chain fatty acids (SCFAs) and neuroinflammation in the section "Effects of gut microbiota on SCFA regulation."

Thank you for pointing out this. we agree with this point. Therefore we have added: 

2.1. Gut microbiota effects on SCFAs regulation

The most abundant SCFAs detected in the faecal are acetate, butyrate, and propionate within the GIT [33]. Butyrate is mainly produced by Firmicutes including Lachnospiraceae and Faecalibacterium prausnitzii, while propionate is primarily produced by Bacteroides species, Clostridium species and Negativicutes [34]. SCFAs are essential for modulating the immune system and maintaining intestinal homeostasis. For example, SCFAs exhibit anti-inflammatory effects by modulating immune cell activity, reducing of pro-inflammatory cytokine release and strengthening the integrity of the intestinal barrier to prevent the translocation of endotoxins [35]. Neuroinflammation is a key factor when considering brain function. Studies have shown that disturbances in the gut microbiota such as induced by antibiotics, lead to systemic immune dysregulation, characterised by pro-inflammatory profiles [35]. In the CNS, antibiotic-induced depletion of the microbiota activates an inflammatory response that alters microglial morphology. Microglia as the resident immune cells of the CNS which are responsive to changes in the gut microbiota composition and its metabolites [36]. SCFAs particularly butyrate has emerged as key mediators linking gut microbiota to neuroinflammation [37]. Sodium butyrate has been shown to inhibit microglial activation and the secretion of pro-inflammatory cytokines, thereby reducing neuroinflammation. For instance, studies have demonstrated that sodium butyrate can inhibit lipopolysaccharide-induced depression-like symptoms in mice by modulating microglial activation [38]. Additionally, butyrate has been shown to promote the differentiation of regulatory T cells and reduce the pro-inflammatory cytokines production, further highlighting its anti-inflammatory potential properties [35]. Moreover, SCFAs can affect the GBA axis through multiple mechanisms including modulation of the vagus nerve and alteration of gut barrier function, and direct effects on NTs production. These mechanisms contribute to the neuroinflammation regulation and may provide new insights into the treatment of neurodegenerative and neurodevelopmental disorders [39]. (in the manuscript Line 137 to 162. 

Comments 2: The discussion section would benefit from a more in-depth exploration of metabolomics, nutraceuticals, their applications in personalized nutrition, and potential future research directions in the field.

Thank you. We agree with this point and we have added an paragraph: It can be seen that metabolomics is a powerful tool in the field of personalised nu-trition, which providing detailed insights into the metabolic fingerprint of individuals that can guide the development of tailored nutraceutical interventions [197]. By identi-fying specific metabolic pathways affected by malnutrition, metabolomics aids in un-derstanding how nutraceuticals affect metabolic well-being and provide a scientific basis for personalised dietary recommendations. Nutraceuticals include a range of bioactive compounds such as probiotics and prebiotics that have demonstrated their potential in the fight against malnutrition by modulating cellular signaling pathways, reducing oxidative stress and enhancing gut health to improve overall nutritional status [198]. The inte-gration of multi-omics technologies including genomics, microbiomics and metabolomics, presents a future vision where personalised nutraceutical interventions are driven by a comprehensive understanding of an individual’s genetic makeup, gut microbiota composition and metabolic profile. This method has the potential to transform malnu-trition management by offering more targeted and effective treatment strategies. Future research directions should concentrate on deciphering the mechanisms by which nutraceuticals interact with the gut microbiome to shape metabolic health and on de-veloping biomarkers that can predict individual reactions to particular nutraceutical treatments [197]. Additionally, it is imperative to investigate the long-term effects and sustainability of personalised nutraceutical interventions in improving GBA health and combating malnutrition [199]. (in the manuscript Line 771 to 789). 

Reviewer 2 Report

Comments and Suggestions for Authors

The manuscript presents a complex bidirectional communication network between the gastrointestinal tract and the brain, influencing both neurological and physiological functions. The interrelationship of GBA modulation with feeding behavior, immune response and neurophysiological processes is examined; cognitive impairment, appetite dysregulation and impaired immunity.
The authors focus on biomarkers derived from the gut microbiota and GBA dysfunction in malnutrition, while evaluating the evidence-based potential of nutritional supplements, including probiotics, prebiotics, synbiotics, postbiotics and paraprobiotics, in combating malnutrition and improving GBA health.
As a future personalization of the problem, nutraceutical intervention is proposed that tailors treatment to individual genetic and microbiome characteristics, providing a more targeted and effective approach to develop precise formulas and optics
poor health outcomes.
The submitted manuscript is written in standard English, according to the requirements of the journal and completely provides a new approach in clinical practice for preliminary determination of risk categories; for which data from the last 5 years were used. The study is entirely medically focused.
Minor remarks:
1. The proposed abstract is satisfactory, but too informative - to be revised.
2. The proposed introduction is accessible and sufficient. To increase the readability of the proposed work, a table has been introduced.
3. Materials and methods are well systematized and described.
Figures 1 and 2 are described and introduced in detail. Figure 3 - to supplement the descriptive part in detail.
The proposed table 3 is too large and the reader loses the connection. I recommend separation, according to the authors' criteria; or modification according to a given factor.
4. Results
Is it possible to introduce schemes showing the specific role of pro-, pre-, sym-,..... biotics in subparagraphs 3.1, 3.2, 3.3, 3.4, in order to increase readability.
The second part of the proposed manuscript presents interesting information regarding in vitro and in vivo approaches to restoring the vai connection.
Paragraphs 541- 552 should be shortened...
5. The conclusion is good.
6. The proposed literature presents new research on the proposed issue within 37% of research from the last 5 years. In places, the requirements of the list have not been met - all technical errors should be removed.

Author Response

Dear reviewer, thank you so much for your kindness. Your comments are quite helpful for us. Based on your comments, We have corrected the manuscript. 

Comment 1. The proposed abstract is satisfactory, but too informative - to be revised.

Thank you for pointing this and we agree with this. We have made a brief abstract: The gut-brain axis (GBA) is a bidirectional communication network between the gastrointestinal tract and the brain, modulated by gut microbiota and related biomarkers. Malnutrition disrupts GBA homeostasis, exacerbating GBA dysfunction through gut dysbiosis, impaired neuroactive metabolite production and systemic inflammation. Nutraceuticals including probiotics, prebiotics, synbiotics, postbiotics and paraprobiotics, offer a promising approach to improving GBA homeostasis by modulating gut microbiota composition and related neuroactive metabolites. This review aims to elucidate the interplay between gut microbiota-derived biomarkers and GBA dysfunction in malnutrition and evaluate the potential of nutraceuticals in combating malnutrition. Furthermore, it explores the future of personalised nutraceutical interventions tailored to individual genetic and microbiome profiles, providing a targeted approach to optimise health outcomes. The integration of nutraceuticals into GBA health management could transform malnutrition treatment and improve cognitive and metabolic health. (In the manuscript line 14 to 25). 

Comments 2: Figure 3 - to supplement the descriptive part in detail.

Thank you for pointing this and we agree with this. we have added: Figure 3. Process of a tailored nutraceutical intervention. This figure illustrates the process of a tailored nutraceutical intervention, which is a personalised approach to improve health outcomes. The process is divided into five main steps including Initial Assessment, Data Analysis, Tailored Personalised Plan Design, Implementation of Intervention and Monitoring and Evaluation, de-signed to optimise health outcomes through personalised nutraceutical interventions. Each step is crucial for ensuring that the intervention is both effective and tailored to the unique needs of the individual. (In the manuscript line 643 to 649). 

Comments 3: The proposed table 3 is too large and the reader loses the connection. I recommend separation, according to the authors' criteria; or modification according to a given factor.

Thank you for pointing this. we have classified the table into five categories including probiotic, prebiotics, synbiotics, postbiotics and paraprobiotics (as seen in the manuscript table 3). 

Comments 4: Results: Is it possible to introduce schemes showing the specific role of pro-, pre-, sym-,..... biotics in subparagraphs 3.1, 3.2, 3.3, 3.4, in order to increase readability.

Thank you pointing this and we agree with this. we have added subtitles in the manuscript including (1) 3.1.1. Effect of probiotics on gut microbiota composition and NTs modulation (line 328). (2)3.2.1. Effect of prebiotic type on neuroactive metabolites (line 375). (3). 3.4.1. Effect of postbiotics on appetite regulation (line 457). (4). 3.5.1. Effect of paraprobiotics on neuroactive compounds regulation (line 493). 

Comments 5. Paragraphs 541- 552 should be shortened. 

Yes we agree with your point and we have revised it: The complexity of the human digestive system challenges experimental replication and affects the accuracy of microbial community dynamics studies. The absence of multiple tissue interactions and metabolites accumulation creates an artificial gut environment. This results in the “bottle effect,” in which the ecological characteristics of the microbial community in vitro start to differ from the actual situation in vivo [160]. However, ad-vancements in culture techniques, tissue cultures, and the establishment of diverse gut microbiota libraries such as the Global Microbiome Conservancy, have significantly enhanced the scope of in vitro research. Moreover, studies integrating host tissues also improve the relevance of these systems by incorporating diverse human-derived strains and host-microbe crosstalk [162]. (in the manuscript line 571 to 580). 

Reviewer 3 Report

Comments and Suggestions for Authors

Main Aspects

The review addresses a topic with growing scientific interest: the relationship between nutraceuticals and the gut-brain axis (GBA) as a therapeutic tool to tackle malnutrition. It is an ambitious and timely approach, which benefits from the integration of different classes of bioactive compounds in a narrative that combines physiological mechanisms, gut microbiota, and neurocognitive effects.

However, several limitations can be found that should be considered before possible publication. Below is an evaluation based on the SANRA tool (Scale for the Assessment of Narrative Review Articles), as well as specific recommendations to improve the scientific, structural, and linguistic quality of the manuscript.

SANRA Score Evaluation

  1. Justification of the importance of the topic

The topic is clearly relevant, but the introduction should present a stronger justification, including recent epidemiological data on malnutrition and its impact on the nervous system. It would also be helpful to highlight why nutraceuticals represent an especially promising approach compared to other therapeutic strategies.

As a suggestion, I recommend that the authors include more recent data on malnutrition prevalence and its relationship with neuropsychological alterations, as well as a better justification for choosing the gut-brain axis as a therapeutic target.

  1. Clarity in the formulation of objectives

The manuscript's objectives are mentioned implicitly but are not explicitly stated as a clear question or hypothesis. This makes it difficult to understand the specific focus of the review.

The authors should include a clear sentence in the introduction or a specific paragraph under the title “Objectives” that states what they intend to analyze, compare, or synthesize. They should also divide the objectives into concrete subtopics: effects of nutraceuticals on GBA, involved mechanisms, human clinical studies, etc.

  1. Adequate description of relevant literature

The article covers a wide range of bioactive compounds, including omega-3s, polyphenols, vitamins, probiotics, and peptides, which adds richness to the content. However, it lacks a critical approach: limitations, contradictions, or gaps in the literature are barely discussed.

The article should include a section with studies showing unfavorable or controversial results to provide a more balanced view and to point out the methodological limitations of all the reviewed studies.

On the other hand, no search method is described, nor inclusion/exclusion criteria, nor the databases used. This is acceptable in a narrative review, but it limits the transparency of the work. The authors should indicate what type of studies (trials, reviews, etc.) they considered, which databases they searched, the approximate time frame, and the quality criteria used for article selection.

  1. Logical and structured presentation of topics

The thematic organization is appropriate and follows a reasonable scheme by groups of compounds. Even so, some sections are overly extended or deviate from the focus on the gut-brain axis, making the manuscript less coherent.

Shorten and better focus some subsections, especially those that repeat information or stray from the main topic. Use transition phrases to facilitate reading.

5. Adequate presentation of relevant data

Multiple studies are mentioned, but key data (e.g., sample size, study type, dose, duration, statistical significance) are often not specified, making it difficult to assess the quality of the evidence.

When studies are mentioned, it would be helpful for the authors to include specific data such as number of participants, the supplement used, the time period, etc. It should also be specified whether the results are from preclinical studies in animals or in vitro.

6. Other aspects

It is advisable for the authors to use a tool such as ROBIS to assess potential risk of bias and report the results obtained for greater transparency. In addition, it is recommended to evaluate the certainty of the evidence using GRADE.

Author Response

Dear reviewer, thank you very much for your great support. Your comments are really helpful for us. Based on your comments, we have corrected the manuscript. 

Comments 1.  Justification of the importance of the topic
The topic is clearly relevant, but the introduction should present a stronger justification, including recent epidemiological data on malnutrition and its impact on the nervous system. It would also be helpful to highlight why nutraceuticals represent an especially promising approach compared to other therapeutic strategies.

Thank you for pointing this and we agree with this. We have added : The double burden of malnutrition includes obesity/overweight and overnutrition [5]. The worldwide impact of malnutrition includes various nutritional issues like un-dernutrition, overnutrition and micronutrient deficiencies, all of which pose major public health concerns. In 2022, approximately 2.5 billion adults were overweight, with 890 million classified as obese and 390 million underweight. Among children under five, 149 million were stunted, 45 million were wasted, and 37 million were overweight or obese (WHO, 2024). Malnutrition affects not only associated with physical health consequences but also has profound impacts on the nervous system, leading to cognitive impairments, mood disorders, and developmental delays (Merino et al., 2024). For example, unbal-anced intake of essential nutrients can lead to impaired gut barrier function, weakened immune function, and altered levels of neurotransmitters (NTs) and appetite hormones, ultimately affecting brain health and cognitive performance. and further exacerbating GBA dysregulation. Hence, combating malnutrition and re-establishing equilibrium in the GBA are vital for optimising health outcomes and mitigating associated comorbidities. The prevalence of malnutrition and its neuropsychological impacts highlights the need for effective therapeutic strategies. Nutraceuticals represent a particularly promising approach compared to conventional therapeutic strategies due to their multiple benefits and minimal side effects. Compared to pharmaceutical drugs, nutraceuticals are derived from natural sources and often have a lower risk of adverse effects. Moreover, nutraceuticals can target multiple pathways such as modulating gut microbiota composition and its metabolites, reducing inflammation and enhancing gut barrier integrity, which are all crucial for improving GBA health [Elazzazy et al., 2025]. (In the manuscript line 49 to 70). 

Comments 2. 2.    Clarity in the formulation of objectives
The manuscript's objectives are mentioned implicitly but are not explicitly stated as a clear question or hypothesis. This makes it difficult to understand the specific focus of the review.
The authors should include a clear sentence in the introduction or a specific paragraph under the title “Objectives” that states what they intend to analyze, compare, or synthesize. They should also divide the objectives into concrete subtopics: effects of nutraceuticals on GBA, involved mechanisms, human clinical studies, etc.

Thank you so much for your advice that is really helpful. We have added: This review addresses three central questions (1). How do malnutrition-induced disrup-tions in the GBA drive neurological and metabolic dysfunction? (2). What is the impact of nutraceuticals on gut microbiota composition and GBA-related biomarkers in the context of malnutrition? (3). What are the prospects and challenges for developing personalised nutraceutical interventions based on individual genetic and microbiome profiles? To answer these questions, this review aims to explore the mechanisms underlying the GBA in malnutrition, the potential of nutraceuticals in improving gut microbiota composition and GBA related biomarkers, and the prospects for personalised nutraceutical inter-ventions. By highlighting the research findings and identifying gaps in current knowledge, this review seeks to provide insights into the future directions of tailored nutraceutical research and its potential applications in addressing malnutrition.(In the manuscript line 83 to 94). 

Comments 3: The article covers a wide range of bioactive compounds, including omega-3s, polyphenols, vitamins, probiotics, and peptides, which adds richness to the content. However, it lacks a critical approach: limitations, contradictions, or gaps in the literature are barely discussed.
The article should include a section with studies showing unfavorable or controversial results to provide a more balanced view and to point out the methodological limitations of all the reviewed studies.
On the other hand, no search method is described, nor inclusion/exclusion criteria, nor the databases used. This is acceptable in a narrative review, but it limits the transparency of the work. The authors should indicate what type of studies (trials, reviews, etc.) they considered, which databases they searched, the approximate time frame, and the quality criteria used for article selection.

Thank you for pointing this. we have added: Although it is widely accepted that nutraceuticals including probiotics, prebiotics, synbiotics, postbiotics and paraprobiotics have the potential to enhance gut health by modulating gut microbiota composition and improving GBA-related biomarkers, this perspective is oversimplistic and ignores the complexity of individual responses. The assumption that these interventions are generally effective for addressing malnutrition is problematic because it does not account for the significant variability in gut microbiota composition and GBA-related biomarkers among individuals with different types of malnutrition. In reality, the efficacy of nutraceutical interventions is highly dependent on individual genetic and microbiome profiles, which can vary widely even within the same category of malnutrition. This variability implies that a generic approach to nutraceutical supplementation is unlikely to produce consistent or optimal results. Instead, precision nutrition that takes into account the specific nutritional needs and gut microbiome characteristics of an individuals is essential for choosing the most appropriate nutraceuticals. However, even precision nutrition faces challenges of accurately categorising and addressing various types of malnutrition, highlighting the need for more nuanced and personalised strategies.

Comments 4. The thematic organization is appropriate and follows a reasonable scheme by groups of compounds. Even so, some sections are overly extended or deviate from the focus on the gut-brain axis, making the manuscript less coherent.
Shorten and better focus some subsections, especially those that repeat information or stray from the main topic. Use transition phrases to facilitate reading.

Thank you for pointing this. We have added some subtitles to make the manuscript more clearly including: (1) 3.1.1. Effect of probiotics on gut microbiota composition and NTs modulation (line 328). (2)3.2.1. Effect of prebiotic type on neuroactive metabolites (line 375). (3). 3.4.1. Effect of postbiotics on appetite regulation (line 457). (4). 3.5.1. Effect of paraprobiotics on neuroactive compounds regulation (line 493).

Comments 5 and 6: 5. Adequate presentation of relevant data
Multiple studies are mentioned, but key data (e.g., sample size, study type, dose, duration, statistical significance) are often not specified, making it difficult to assess the quality of the evidence.
When studies are mentioned, it would be helpful for the authors to include specific data such as number of participants, the supplement used, the time period, etc. It should also be specified whether the results are from preclinical studies in animals or in vitro.
6. Other aspects
It is advisable for the authors to use a tool such as ROBIS to assess potential risk of bias and report the results obtained for greater transparency. In addition, it is recommended to evaluate the certainty of the evidence using GRADE.

We appreciate the reviewer’s insightful comment regarding the review
type. Upon careful consideration, we believe this manuscript aligns more
closely with a comprehensive review—a broad synthesis of existing
literature on [Nutraceuticals for Gut-Brain Axis Health: A Novel Approach to Combat Malnutrition and Future Personalised Nutraceutical Interventions]—rather than a narrative review. While narrative reviews often employ selective inclusion criteria and require explicit database/search methodology, comprehensive reviews aim for exhaustive coverage of key developments without mandated database declarations.